# Semi-Automatic Methodology for Fire Break Maintenance Operations Detection with Sentinel-2 Imagery and Artificial Neural Network

**João E. Pereira-Pires** [1,*] , **Valentine Aubard** [2] , **Rita A. Ribeiro** [1] , **José M. Fonseca** [1] ,
**João M. N. Silva** [2] **and André Mora** [1]

[1] Centre of Technology and Systems/UNINOVA, School of Science and Technology–NOVA University of Lisbon, 2829-516 Caparica, Portugal; rar@uninova.pt (R.A.R.); jmf@uninova.pt (J.M.F.); atm@uninova.pt (A.M.)

[2] Forest Research Centre, School of Agriculture–University of Lisbon, 1349-017 Lisbon, Portugal; vaubard@isa.ulisboa.pt (V.A.); joaosilva@isa.ulisboa.pt (J.M.N.S.)

\* Correspondence: je.pires@campus.fct.unl.pt; Tel.: +351-212948527

**Abstract:** The difficult job of fighting fires and the nearly impossible task to stop a wildfire without great casualties requires an imperative implementation of proactive strategies. These strategies must decrease the number of fires, the burnt area and create better conditions for the firefighting. In this line of action, the Portuguese Institute of Nature and Forest Conservation defined a fire break network (FBN), which helps controlling wildfires. However, these fire breaks are efficient only if they are correctly maintained, which should be ensured by the local authorities and requires verification from the national authorities. This is a fastidious task since they have a large network of thousands of hectares to monitor over a full year. With the increasing quality and frequency of the Earth Observation Satellite imagery with Sentinel-2 and the definition of the FBN, a semi-automatic remote sensing methodology is proposed in this article for the detection of maintenance operations in a fire break. The proposed methodology is based on a time-series analysis, an object-based classification and a change detection process. The change detection is ensured by an artificial neural network, with reflectance bands and spectral indices as features. Additionally, an analysis of several bands and spectral indices is presented to show the behaviour of the data during a full year and in the presence of a maintenance operation. The proposed methodology achieved a relative error lower than 4% and a recall higher than 75% on the detection of maintenance operations.

**Keywords:** remote sensing; fire break; object-based classification; change detection; wildfires; artificial neural networks; sentinel-2

## 1. Introduction

Wildfires are among the most destructive disasters. These catastrophes have an enormous impact in populated regions. It applies to the United States (Western states) [1]; Canada (South-Western) [2]; Mediterranean Europe (Portugal, Spain, France, Italy and Greece) [3] and South Eastern Australia [4,5]. As can be seen, this is a worldwide problem and not just on less developed countries. The means available for fighting the wildfires are clearly insufficient for their efficient suppression. The answer to this problem must be not only reactive, but also proactive. The prevention of fires and the implementation of strategies to help the firefighting are imperative.

One of the possible strategies is the implementation of a fire break network (FBN). A fire break (FB) is a strip of land that has been strategically and artificially modified, where vegetation density is reduced to break up the continuity of fuel. It acts as a barrier to slow or stop the progress of wildfire,

thus improving fire control opportunities. Technically, in the Portuguese FBN, an FB is a land strip with 125 m wide composed by three regions: the road network, with a minimum width of 5 m; the fuel interruption with a minimum width of 10 m where all vegetation is cut; the fuel reduction composed by two zones, where a minimum distance between the tree tops is imposed (the complete information is available in [6]). The technical specifications are synthetized in Figure 1 and a ground observation of an FB can be seen in Figure 2.

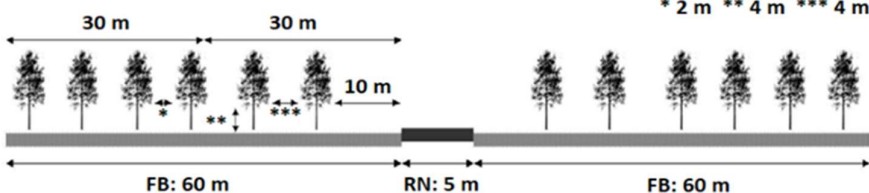

**Figure 1.** Technical specifications of a fire break (FB) in the Portuguese fire break network (FBN) [6].

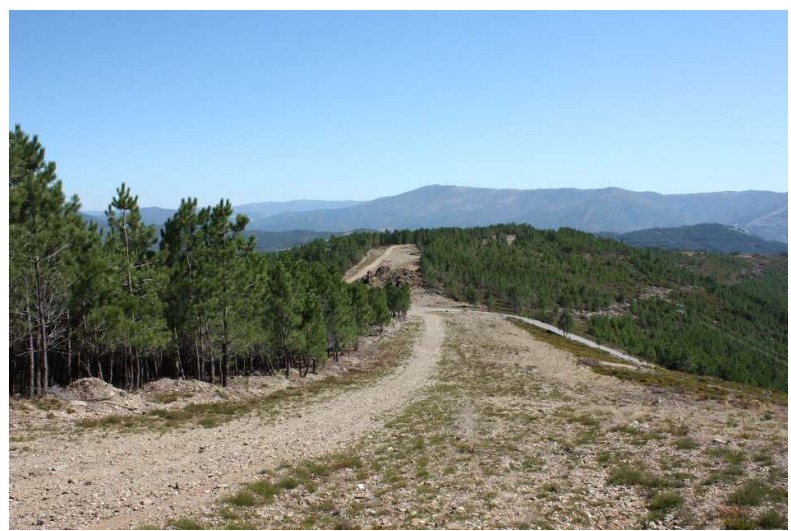

**Figure 2.** Ground observation of an FB. (Source: http://www2.icnf.pt/).

Since the vegetation is always growing, the monitorization of an FB is essential for its efficiency. The Portuguese Institute of Nature and Forest Conservation (Instituto de Conservação da Natureza e das Florestas, ICNF) planned a priority network of FBs to help control wildfires and decrease their burnt area. The implementation and maintenance of FB is crucial for its efficiency and should be ensured by the Local Authorities. The verification by the National Authorities (ICNF) of the previous premises can be made by ground observation, but it is expensive and time consuming if a wide network is implemented (the FBN was defined as having 11,125 km, having 1600 km already implemented). The current practice is the visualization of satellite imagery, pointing out the cases that need special attention. This technique is also a time-consuming process and prone to errors. In this paper, a remote sensing semi-automatic methodology for the detection of maintenance operations in an FB is presented.

With the launch of the Sentinel 2 (the first satellite of the constellation in June 2015 and the second in March 2017) free satellite imagery with a spatial resolution of 10 m and a temporal resolution of five days was made available to the community. This new high spatial resolution allowed the analysis of the FB conditions, which in current work, was to detect when maintenance operations are performed. It is common to divide remote sensing applications into two groups: land cover classification [7–10] and change detection [11–17]. The proposed methodology fits into the second group. In this kind of applications, the output gives the information of the occurrence of an event in the study area. In Hamunyela et al. [12], forest disturbances were detected with resource to two observations and spatio-temporal features and in Hermosilla et al. [13]; annual composites were

generated to detect changes. In [15–17], different approaches were used to identify changes in the land cover and the kind of changes. In [16,17], pixel-based methods were implemented, with three and six observations, respectively, while in [15], an object-based technique was used. It should be noticed that in these works [11–17] Landsat imagery was used, which means that the best achievable detection period is 16 days (but due to the atmospheric conditions the period is usually worse). Additionally, with the exception of [15], more than one observation of previous data is needed for the detection. Usually, the change detection relies on the slope of the time-series and consecutive variations in vegetation indices. These applications showed the importance of the temporal and spatial resources for disturbances detection.

The purpose of this work is to identify when maintenance operations occur in the FBs. Although there is not an evaluation of the quality of the operation, only complete operation in the FB are to be detected. Relatively to the common change detections methods the goals were:

- Use of Sentinel-2 data instead of Landsat imagery, due to its increased frequency and spatial resolution;
- Identify only a specific kind of operation efficiently, dealing with the phenology and different types of vegetation;
- Use of common vegetation indices and other indices;
- Reduce the previous data used, identifying the maintenance as soon as possible, allowing a classification whenever a new observation is made, as in [15].

To achieve these objectives and based on current literature, several requirements were defined as follows:

- Object-based classification—since an FB is a well-defined area, it will be defined as an object. This approach can—better capture its spatial characteristics;
- Temporal dynamics—the use of time-series allows the determination of the temporal dynamics, which is essential in change detection methods;
- Machine learning—the use of artificial intelligence techniques to enhance the change detection classification.

There are two main stages: the data extraction, and the maintenance operations detection with an artificial neural network (ANN). In the first stage, the time-series and the datasets are created. It includes the geolocation correction of the observations, calculation of the mean values of the reflectance bands within the FB, followed by the application of a noise reduction filter and finally, computation of the spectral indices. The second stage includes a preliminary training step for feature selection, ANN training and error estimation, and a classification step to identify the months where a maintenance operation was executed.

The structure of this paper is as follows: Section 2 describes the study area, the data used for the implementation, and presents the dataset and the detection methodology; Section 3 show the results of the intermediary steps for designing the ANN and the detection results followed by their discussion in Section 4. Finally, the conclusions are in Section 5.

## 2. Materials and Methods

### 2.1. Study Definition

In this work, four study areas (Fundão, Seia, Serra dos Candeeiros and Sertã) within the Portuguese FBN were defined. These had an FB installed or operated during the study period, which was 2017 and 2018 (first two years with both sentinel-2 A and B satellites). Beyond the defined FBs, several vegetation areas (VEG), near the study cases, were analysed for behaviour comparison and ANN classifier training. Another study area was identified (Marisol) to test the ANN. This FB does not belong to the FBN, but to the Power Line Network, and an operation was executed during the study

period. In Figure 3, observations are presented of an FB before and after a maintenance operation. These disturbances in the land are observable with the naked eye.

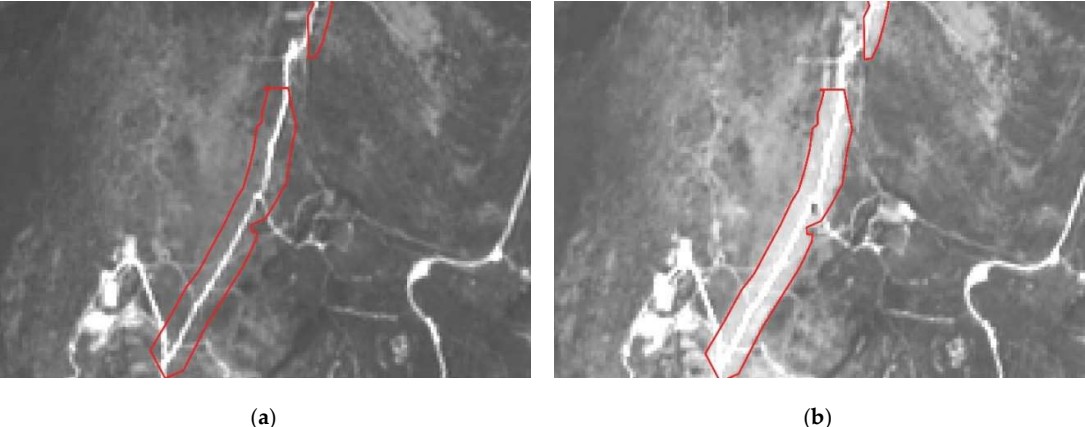

(**a**)　　　　　　　　　　　　　　　　　　　　　　　　　　　　　　(**b**)

**Figure 3.** Example of a maintenance operation on an FB (red) in Serra dos Candeeiros, as seen by Sentinel-2 Band04 images. (**a**) before operation (08-05-2017); (**b**) after operation (04-07-2017).

The flora richness is also a Portuguese characteristic, so the definition of several study areas is important for the validation and training of the maintenance detection classifier. According to the Portuguese land-cover use map from 2015 (COS2015), the most up to date map available, vegetation in these areas are:

- Fundão: pinaster and eucalyptus forests and bush areas;
- Marisol: eucalyptus forests;
- Seia: artificial territories and bush areas;
- Serra dos Candeeiros: agricultural zones and bush areas;
- Sertã: bush areas.

*2.2. Materials and Datasets*

The data used were the FB shapefiles, provided by ICNF, and the Sentinel-2 images. The Sentinel-2 images are Level 1C (L1C) products, since Level 2A (L2A) products only became available since March of 2018. The L1C imagery is Top-of-Atmosphere radiance of each band, while L2A has an additional correction corresponding to the Bottom-of-Atmosphere radiance.

From the available Sentinel-2, observations only bands with a spatial resolution of 10m and 20m were used. Images with 60m spatial resolution were not used, due to the small width of an FB. The bands analysed were: B02, B03, B04, B05, B07, B08, B8A, B11 and B12. Besides the spectral bands, several vegetation and other spectral indices were calculated; (Table 1): normalized difference moist index (NDMI), normalized difference vegetation index (NDVI), ratio vegetation index (RVI), normalized multi-band drought index (NMDI [18]), normalized difference index (NDI), excess of green (ExG), excess of red (ExR), excess of green minus excess of red (exgr) and modified excess of green (MExG) [19].

The NDVI and NDMI indices have several applications for change detection in the vegetation [20,21]. Additionally, NDI highlights the distinction between vegetation and soil land cover types [22]. The visible spectrum indices ExG, ExR, ExGR and MExG also evidence good results in the separation of vegetation from the background [23,24].

**Table 1.** Spectral indices equations.

| Index | Equation |
|---|---|
| Normalized Difference Moisture Index (NDMI) | $\frac{B08-B11}{B08+B11}$ |
| Normalized Difference Vegetation Index (NDVI) | $\frac{B08-B04}{B08+B04}$ |
| Ratio Vegetation Index (RVI) | $\frac{B04}{B08}$ |
| Normalized Multi-Band Drought Index (NMDI) | $\frac{B8A-(B11-B12)}{B8A+(B11-B12)}$ |
| Normalized Difference Index (NDI) | $128\times\left(\frac{B03-B04}{B03+B04}+1\right)$ |
| Excess of Green (ExG) | $2\times B03-B04-B02$ |
| Excess of Red (ExR) | $1.3\times B04-B03$ |
| Excess of Green minus Excess of Red (ExGR) | $ExG-ExR$ |
| Modified Excess of Green (MExG) | $0.441\times B04-0.811\times B03+0.383\times B02+18.78745$ |

The vegetation has a typical dynamic throughout the whole year. Due to phenology, the spectral reflectance of the vegetation is not constant. There are some trends that can be observed in the different seasons of the year, particularly in the summer and in the winter. The influence of these cyclical effects in vegetation areas is shown in Figure 4. Note that Portugal has a Mediterranean climate, which implies four well defined seasons. This enhances phenology dynamics, a drawback for the detection of the maintenance operations. This is because a change detection methodology looks for variations in data, so it is important to understand if the disturbance occurs due to phenology or a maintenance operation. In Figure 5 are presented two vegetation indices time-series where the two kinds of transitions are shown. It may also be concluded that there are indices more robust to the phenology, for instance, in Figure 5b with ExG, the operation is more evident.

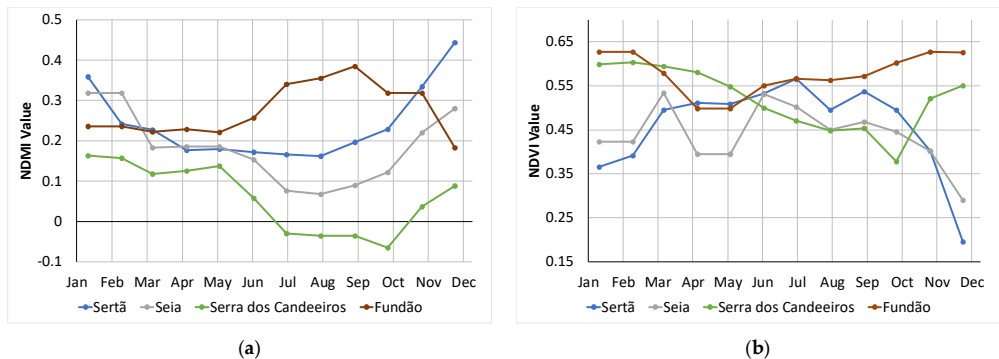

(**a**)　　　　　　　　　　　　　　　　　(**b**)

**Figure 4.** Spectral indices time-series from vegetation areas during 2017 for each study area. Here, both indices are strongly affected by phenology. (**a**) NDMI; (**b**) NDVI.

Due to cloudiness and snow (Fundão and Seia) some of the observations were not clear, winter months being the most affected ones. In Table 2, the number of clear observations per study area and year, number of FBs and the months defined for the operations are shown. From the study areas, a total of 14 FB and 9 VEG areas were defined. The maintenance operation or implementation dates were given by ICNF and confirmed by visual inspection of Sentinel-2 imagery.

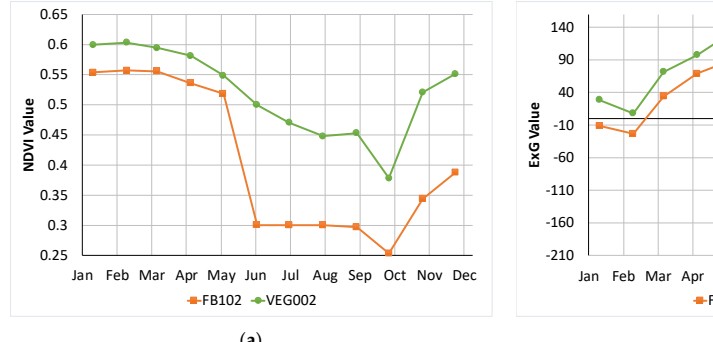
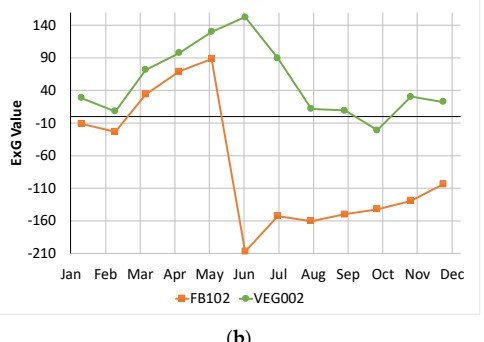

(**a**)  (**b**)

**Figure 5.** Spectral indices time-series comparison from FB and VEG areas in Serra dos Candeeiros during 2017. Phenology effects are present in both indices, but when an operation occurs, ExG is more robust than NDVI. (**a**) NDVI; (**b**) ExG.

**Table 2.** Number of clear observations per study area and year.

| Study Area | Area [ha] | Clear Observations | | Number of FBs | Operation Dates |
|---|---|---|---|---|---|
| | | **2017** | **2018** | | |
| Fundão | 71.2 | 28 | 23 | 3 | JUL/AUG/SEP 2017 |
| Marisol | 2.0 | 40 | 49 | 1 | MAR 2018 |
| Seia | 50.2 | 29 | 24 | 2 | JUN 2017 |
| Serra dos Candeeiros | 105.5 | 59 | 49 | 5 | MAY/JUN/JUL/AUG 2017 |
| Sertã | 53.2 | 30 | 19 | 3 | JAN/FEB/JUN 2017 JUL 2018 |

As can be seen in Figure 6, some of the maintenance operations occur in two successive months. This fact leads to the definition of two kinds of operations: the instantaneous operations, that occur in one month only, and the continuous operations. Normally, the latter start at the end of the month and continue into the beginning of the next month. Note that an operation is not an instantaneous procedure; it takes several days between the vegetation cut and the cleaning of the site (and sometimes it can be noticed between observations). So, it may be executed during the same month or in two consecutive months. In Figure 6, the difference between an instantaneous and a continuous operation is shown. The first is characterized by a single event (June of 2017), while the second by two events (May and June of 2017). The change in a continuous operation is, usually, softer than the instantaneous one, being usually harder to detect.

The implementation of the methodology relied on three datasets: training, validation and test. The first two resort only to data from Fundão, Seia, Serra dos Candeeiros and Sertã. A random stratified split was executed, generating the training and validation dataset (the first with 67% of the data and the second with the remaining). The test dataset is composed of two groups: the real case scenario and the incomplete operations set. The first group is composed by Marisol data, it tested the behaviour of the ANN in an untrained scenario. For the second group, since there is no information about the occurrence of incomplete operations, it was needed to simulate these events. As said before, the maintenance procedures are not instantaneous, which allowed the creation of datasets with partial operations. For this purpose, some observations were removed, being presented to the ANN as an incomplete operation, instead of the real complete operation. In Figure 7, there are several observations of an FB, between (a), (b) and (c), some work is done, but it is only completed in (d). In the real case all observations are used, but for the simulated incomplete operations, each transition is evaluated. The incomplete operations should be divided into different groups, according to their completeness degree. This measure corresponds to the ratio between the area already operated and the FB total area. In Figure 7, an example from Marisol is presented, where in (b), the completeness of the

operation is 28% and in (c) it is 61%. If the maintenance covers 75% or more of the FB, it is considered a complete operation.

The training and validation dataset were composed by 261 observations and for the test dataset there were 89 observations for Marisol and 21 observations for the definition of 9 incomplete operations.

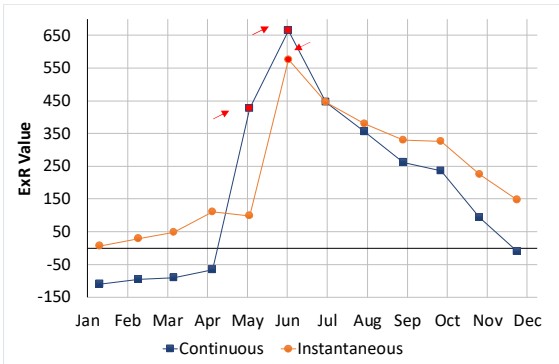

**Figure 6.** Comparison between an instantaneous and a continuous operation in Serra dos Candeeiros during 2017. Operation dates highlighted in red squares and arrows.

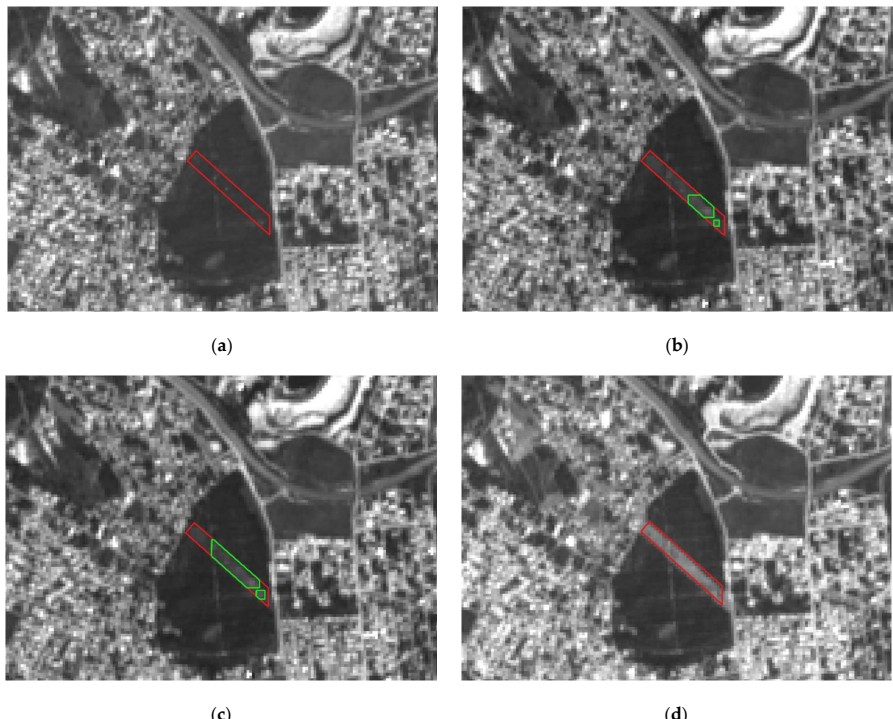

**Figure 7.** Sentinel-2 Band 04 images from an FB maintenance operation in Marisol (in red the total area, in green the operated area): (**a**) before (30-01-2018), (**b**) completeness of 28% (22-02-2018), (**c**) completeness of 61% (24-02-2018) and (**d**) completed (26-03-2018).

*2.3. Data Extraction*

The data extraction stage goals are to obtain for the FB and VEG areas: the time-series from the Sentinel-2 observations, to understand their trends; and the datasets for the ANN training and detection steps. The data extraction procedure is synthetized in Figure 8. The GIS tool used in this study for the methodology implementation was Quantum GIS (QGIS) because it is an open source software and it allows the automation of tasks with the available Python console.

The procedure starts by loading Sentinel-2 data into QGIS, followed by clipping the area of interest (to save time in the geolocation corrections). The geolocation errors are calculated, and the corrections applied to the Sentinel-2 data. The final step is the generation of the time-series and the datasets.

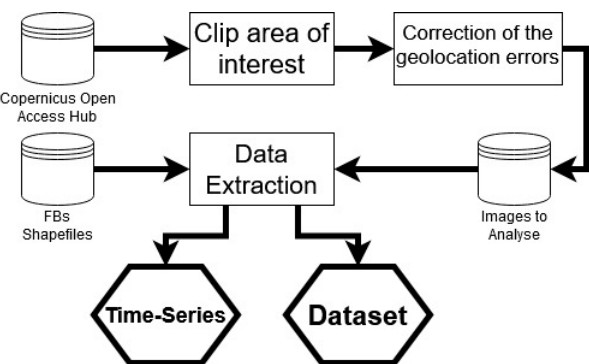

**Figure 8.** Diagram of the data extraction stage.

### 2.3.1. Geolocation Correction

As reported by ESA in [25], the Sentinel-2 L1C products may have a geolocation offset error, even after performing correction procedures [26]. This is only a translational offset and rarely exceeds 1.5 pixels (15 meters), being usually at the sub-pixel level. Normally this error is negligible, because the areas in study are wide enough not to influence the results. In our case, due to the dimensions of the FB (125 m wide) and the spatial resolution of Sentinel-2 (10 m), there are few pixels to use for the FB object representation. So, on the presence of a geolocation error, pixels within the boundaries of an FB shapefile may correspond to vegetation. Additionally, pixels in the other boundary corresponding to the FB will be discarded.

The geolocation correction method used was proposed in [27]. It detects subpixel deviations and is processing efficient. Due to the large amount of information on each observation and to speed up the offset calculations, a smaller area is clipped and used for the analysis, significantly reducing the computation time.

Since the methodology is implemented in QGIS, the geographic coordinates of each pixel play an important role in this study. All the observations are compared with a reference image (the one used for the FB shapefiles design) and the deviations are registered. The algorithm starts with the computation of the maximum value of the cross-correlation between the images, using the Fast Fourier Transform. Then, to detect the subpixel errors, a refinement is made by up-sampling a just narrower region of interest and again the algorithm computes the cross-correlation on a neighbour of 1.5 pixels.

The application of the corrections is done using the GDAL library from OSGeo project (https: //gdal.org/). The images' coordinates are then adjusted correcting the geolocation of the pixels accordingly to the estimated values. After that, the observations are ready for data extraction. Figure 9 shows an example of a geolocation error, being (a) the reference image and (b) an observation of the same site in a different day with a computed geolocation error of 0.7 (7 m) and −0.11 (1.1 m) in North/South and East/West directions, respectively. This is highlighted by the dashed green ellipsis in Figure 9b.

To estimate the error of the geolocation method, two tests were made. The first (*T*1) applies to artificial geolocation errors, allowing an error estimation without uncertainty. The second (*T*2) tests a real scenario, where a set of images was compared with a reference and the uncertainty calculated.

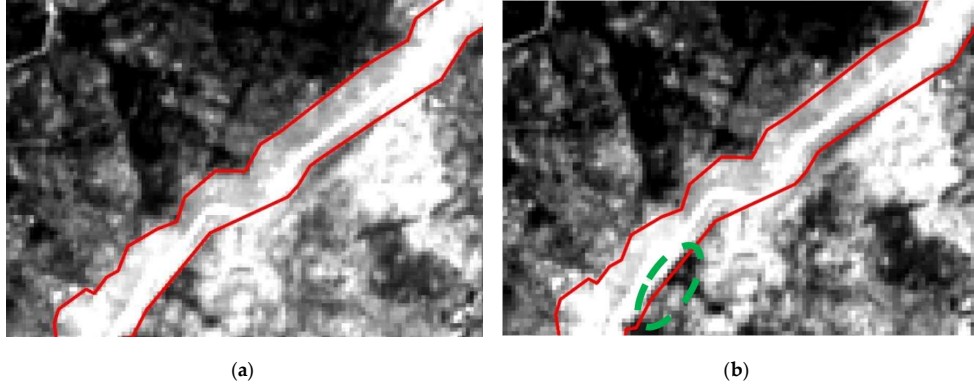

(a)  (b)

**Figure 9.** Geolocation correction in Band 04 images from Seia: (**a**) reference image (18-08-2017); (**b**) corrected image with an estimated offset of 0.7 North/South and −0.11 East/West (24-07-2017).

In $T1$ a set of images were randomly deviated between −1.5 and 1.5 pixels in both directions, creating an artificial geolocation error. The proposed algorithm for the geolocation error estimation compared the deviated observation with the corresponding non-deviated observation. The normalized root mean square error (*NRMSE*), presented in Equation (1), between the calculated geolocation errors and the artificial geolocation errors generated was used as an evaluation metric of this algorithm. In Equation (1) *RMSE* stands for root mean square error, (2), where $y_p$ corresponds to the predicted values and $T$ the number of samples.

$$NRMSE = \frac{RMSE}{y_{max} - y_{min}} \tag{1}$$

$$RMSE = \sqrt{\frac{\sum_{t=1}^{T}(y - y_p)^2}{T}} \tag{2}$$

$T2$ consisted of a comparison of a set of reference images with their corrected geolocation. Then, a smaller area from the corrected observation is clipped and geolocation errors are computed again. The new calculated values correspond to the uncertainty of the method. The error was estimated with the same metric of $T1$, and the results are presented in Table 3. The results show that the maximum error, obtained in the worst-case scenario of 15m deviation, will be less than 1.5 m, which is acceptable.

**Table 3.** Results of the Geolocation Correction.

| Test | NRMSE |
|------|-------|
| T1   | 9%    |
| T2   | 4%    |

### 2.3.2. Image Data Extraction

The data extraction stage is responsible for extracting monthly values for the analysed bands and vegetation indices from the multispectral images, for each FB or VEG areas. The outputs from this procedure are a time-series (used for the feature selection and visualization of data) and a dataset for the detection and algorithm training.

To achieve these goals, the following steps are performed:

1. Extraction of the pixel values from the FB for each band in analysis and calculation of the mean value for the object representation;
2. Normalization of the band values;
3. Generation of monthly values for the FB or VEG regions;
4. Calculation of the defined spectral indices;
5. Concatenation of the previous month values to each month, to include temporal information.

As the goal is to detect the maintenance operation in an object, step 1 is responsible for their representation by a value. The chosen metric was the mean value. To validate it is a representative measure to describe an FB, the relative standard deviation was calculated [28] and 0.185 was obtained. Since it is much smaller than 1, it is a reasonable approximation [29].

In step 2, the band values are normalized with the min-max technique. The minimum and maximum values corresponding to the radiometric resolution of the bands are [0;4095] and the range of normalization [0;1]. A requirement for the change detection classifier is the definition of a timestep. Additionally, time-series are noisy signals and to deal with this, a mean and a high median filter (when the set have a pair number of elements, instead of being the mean of the median elements, it is the higher element) were applied separately, and a comparison was made. A monthly timestep was defined, because two months without observations is unusual in the chosen study areas. In the case that there is not any clear observation for a month, the previous month's values are used.

Figure 10a demonstrates that mean and median filters achieve the intended noise reduction when applied with a monthly window. Additionally, it is shown that the mean filter smooths the data more than the median filter. It is thus expected that the mean filter decreases the ability for the detection of operations. However, the median filter is more sensible to the noise, as shown in Figure 10b, and can origin more false positives, i.e., the detection of incorrect maintenance operations. In Figure 10b, an example of the performance of the noise reduction filters in an FB in Fundão is shown. It can be observed in the bars plot that when an operation occurs (August), the change in data is greater with a median filter. Although, if the operation occurs in the last observation of the month, the median value will be a value one before the cut, hiding the change being detected in the next month.

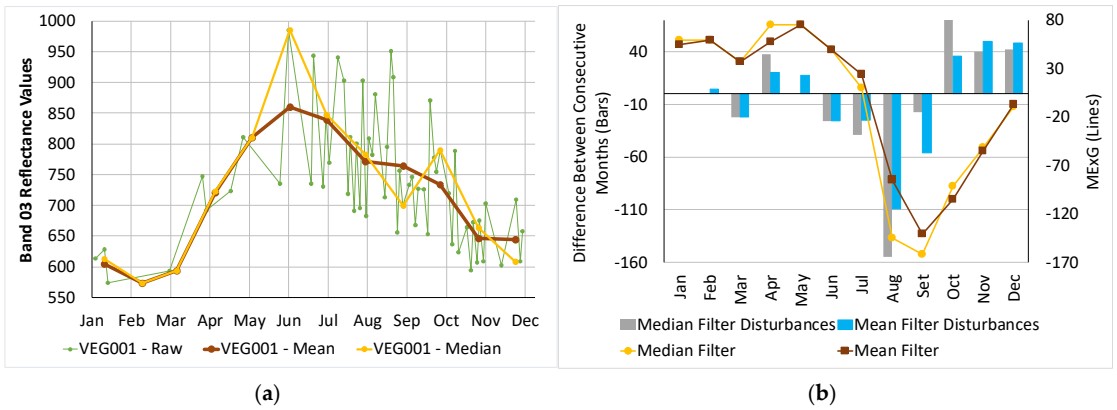

(**a**)                                    (**b**)

**Figure 10.** Comparison of mean and median noise reduction filtering (**a**) Raw and filtered data (band 03, Serra dos Candeeiros in 2017); (**b**) Month disturbances (MExG, Fundão in 2017).

Step 4 begins with the calculation of the spectral indices using the monthly values calculated in the previous step. Alternatively, these could be obtained by calculating the spectral indices per pixel and then aggregating them in a monthly value. However, the first approach saves processing time and does not represent a loss of information that compromises the problem solving. This is proved by Figure 11, where the results for the NDVI using the pixel and object computation with a median filter are presented. The NRMSE between the data obtained from these techniques is 7.4%, and if a mean filter is used, this value is 1.4%, which demonstrates the similarity between the data acquired.

The maintenance detections rely on the temporal disturbances of the data. In step 5, to represent the temporal behaviour in the datasets, the previous month values are concatenated to each data entry.

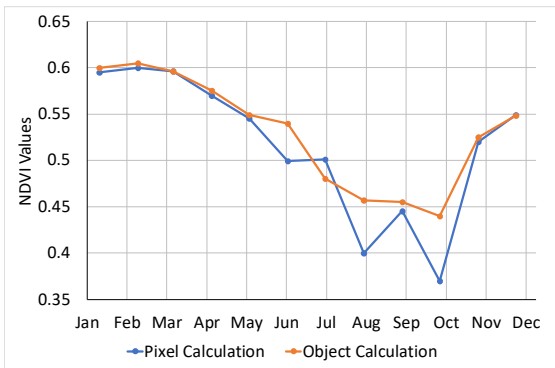

**Figure 11.** Comparison between computation spectral indices using a pixel and object approach.

### 2.4. Maintenance Operations Detection

The algorithm for the maintenance operation detection is represented in Figure 12. The goal is to identify the month where a maintenance operation was undertaken. In this section, the ANN design and training and the detection of operations are discussed.

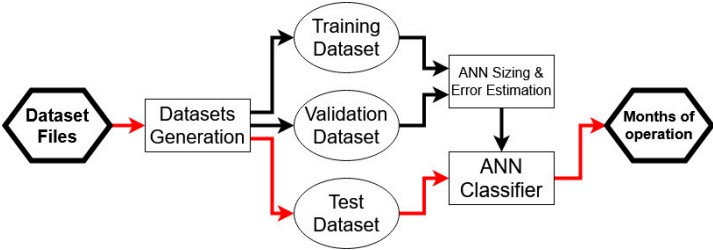

**Figure 12.** Diagram of the maintenance operation detection stage.

2.4.1. Artificial Neural Network Design and Training

The first task is to generate, from the dataset created in Section 2.3, a training and a validation dataset. With this training dataset, the feature selection and the structure of the ANN are defined (number of hidden layers and neurons per layer). The validation set is used to make fine adjustments and to improve the credibility of the error estimation. Since only 3.5% of the training samples correspond to detected operations, it was necessary to add a penalty for the incorrect classification of this class. The Scikit–Learn module does not have this feature, therefore these samples were replicated a number of times equal to the penalty value. In this work a value of 8 was used, increasing the percentage of detected operations to 22.4%.

The feature selection step uses the SelectKBest method (from Scikit–Learn) and the Pearson Correlation over the time-series generated in Section 2.3. The SelectKBest performs a variance analysis and selects the K best features based on a score function. The 10 best features with the highest f_classif score function were selected (this function uses the ANOVA F-Value between label/feature for classification task). Although this analysis chooses the best features, they are selected independently. So, the Pearson Correlation was used to find redundancies and eliminate strong correlated features. After these tasks, multiple sets of features were defined and evaluated. The evaluation consisted of using them to classify the training set and compare their relative errors.

The ANN structure was defined empirically changing the number of neurons and hidden layers pursuing the lowest classification error. The number of neurons per layer chosen was the value that, after increasing, did not improve the ANN. For the number of layers, due to the vanishing gradients problem, the ANN with one and two hidden layers was tested. The input layer uses one neuron per feature and the output layer is only one neuron, since it is a binary class (detection, no detection). The

activation function is the logistic function. The method used for the backpropagation of the error was the stochastic gradient descent, described in [30], with an adaptive learning rate.

### 2.4.2. Training, Validation and Test Error Estimation

The error estimation for the detection methodology was done using the training set, validation set and a test set. For the training, a cross-validation technique was applied. Then, the validation dataset was classified, and relative error, recall, precision and F1-score relativity to both classes, detection and no detection metrics were computed, to assess detection efficiency. Finally, the ANN is applied to the test dataset, evaluating the error on a real case scenario and to assess the robustness to avoid detecting incomplete operations.

## 3. Results

The results will be divided into two sections. On the first, the results that supported the ANN design are presented, while on the second the results of the implemented methodology are presented.

### 3.1. Feature Selection and Artificial Neural Network Sizing

Feature selection techniques were applied to the training set, to reduce the number of features and use the most discriminant ones. This process was applied to both median and mean filtered data. The results presented here will be only for the median filter, due to its similarity with the mean filter. The total number of samples used in the ANN training were 633, which accounted for 67% of all the clear observations in the four study areas multiplied by the 13 FBs.

According to the SelectKBest feature selection algorithm, the best features in this analysis are: B04, B05, B11, B12, NMDI, NDI, ExG, ExR, ExGR, and MExG. Additionally, the Pearson absolute correlations between these features were calculated, to identify which ones give more information together.

Analysing the mean correlation values between a feature and all the others (Table 4), the more independent features are the NMDI and ExG, with 0.751 and 0.748, respectively. The least correlated features pairs were ExG/B05 and ExG/B11, with 0.600 and 0.602, respectively. From this analysis, it was concluded that ExG is probably the most discriminant feature, and consequently it was one of the selected features to be used in the detection. The higher correlated pairs were B11/B12 (0.986) and ExR/ExGR (0.984). In the first pair, the features have a similar behaviour, but B12 has a higher mean correlation, therefore it was rejected. Regarding the other pair, ExR is less correlated to ExG than ExGR, but ExGR works better with B05 and B11 than ExR. The NDI do not enhance the occurrence of a maintenance operation. Finally, B04 is extremely correlated with B05 (0.975), and the same happens with MExG and ExG (0.958). However, with B04 and MExG the same happens as with the NDI.

**Table 4.** Pearson correlation between features.

| | B04 | B05 | B11 | B12 | ExG | ExGR | ExR | MExG | NDI | NMDI |
|---|---|---|---|---|---|---|---|---|---|---|
| **B04** | 1 | 0.975 | 0.929 | 0.939 | 0.731 | 0.931 | 0.973 | 0.845 | 0.946 | 0.702 |
| **B05** | 0.975 | 1 | 0.952 | 0.938 | 0.600 | 0.852 | 0.923 | 0.748 | 0.910 | 0.678 |
| **B11** | 0.929 | 0.952 | 1 | 0.986 | 0.602 | 0.832 | 0.894 | 0.744 | 0.908 | 0.805 |
| **B12** | 0.939 | 0.938 | 0.986 | 1 | 0.648 | 0.857 | 0.909 | 0.773 | 0.927 | 0.794 |
| **ExG** | 0.731 | 0.600 | 0.602 | 0.648 | 1 | 0.922 | 0.840 | 0.958 | 0.789 | 0.670 |
| **ExGR** | 0.931 | 0.852 | 0.832 | 0.857 | 0.922 | 1 | 0.984 | 0.981 | 0.941 | 0.754 |
| **ExR** | 0.973 | 0.923 | 0.894 | 0.909 | 0.840 | 0.984 | 1 | 0.941 | 0.962 | 0.754 |
| **MExG** | 0.845 | 0.748 | 0.744 | 0.773 | 0.958 | 0.981 | 0.941 | 1 | 0.892 | 0.748 |
| **NDI** | 0.946 | 0.910 | 0.908 | 0.927 | 0.789 | 0.941 | 0.962 | 0.892 | 1 | 0.83 |
| **NMDI** | 0.702 | 0.678 | 0.805 | 0.794 | 0.670 | 0.754 | 0.754 | 0.748 | 0.83 | 1 |
| **Mean** | 0.886 | 0.842 | 0.850 | 0.863 | 0.751 | 0.895 | 0.909 | 0.848 | 0.901 | 0.748 |

To access the performance of the classifier for different feature combinations, several groups of features were defined, as presented in Table 5.

**Table 5.** Group of features to evaluate.

| Group | Features |
| :---: | :---: |
| 1 | B05, ExG |
| 2 | B11, ExG |
| 3 | B05, ExG, NMDI |
| 4 | B11, ExG, NMDI |
| 5 | B05, ExG, ExR |
| 6 | B11, ExG, ExR |
| 7 | B05, ExG, ExGR |
| 8 | B11, ExG, ExGR |
| 9 | B05, ExG, ExR, NMDI |
| 10 | B05, ExG, ExGR, NMDI |
| 11 | B11, ExG, ExR, NMDI |
| 12 | B11, ExG, ExGR, NMDI |

First, all the groups were tested on an ANN with one hidden layer, varying the number of neurons in the interval (5,100) with steps of five. In the cross validation, five folds were used and the process was repeated 10 times, with the mean error being used. The results presented in Figure 13a show that is possible to divide the groups into three sets according to the cross-validation error: the first two tend to a relative error of approximately 6%–8%, while on the third set (group 5, 6, 9, 11) the error can be less than 4%. The best results were verified for the second category, i.e., groups 5, 6, 9 and 11. The recall, precision and F1 score for each class (detection and no detection), also demonstrated a better performance for this third set.

Figure 13b shows the recall, used to check the ability to avoid false negatives. Again, groups 5, 6, 9, and 11 were confirmed as better suiting this problem. Although they present similar results, groups 5 and 6 use less features; in these groups NDMI index is not included. If this was used in the data extraction stage, two more bands would be needed (B8A and B12), as well as an extra calculation and two extra neurons in the input layer. This means more processing time, particularly in the data extraction. For these two groups, an ANN with two hidden layers was tested, with the goal of reducing the number of neurons and improve the detection. This test did not show any improvements, so it was defined as an ANN with just one hidden layer.

To decide which of these two groups will be used and the final number of neurons, a final test was done varying the number of neurons in the interval (45,60) (where the error is approximately 3%–5%, and the recall and precision are above 90%), with steps of one neuron. Here, both groups have nearly the same error. Since B05 had better results in the feature selection stage, group 5 was chosen. For the ANN structure, one hidden layer with 53 neurons was defined (with an error lower than 3%), with a penalty of 8. For the mean filtered dataset, the selected features were also from Group 5, and the ANN was composed by an ANN with 53 neurons (the error is approximately 3%), with a penalty of 8. In both, during the training stage and structure definition, a penalty of 10 was used. After that, the penalty was decreased until the error metrics started deteriorating.

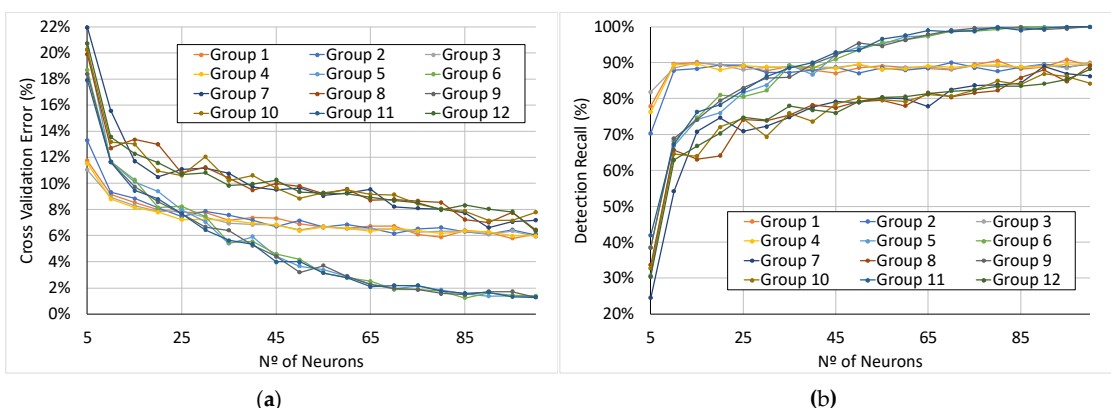

**Figure 13.** Artificial Neural Network (ANN) cross validation results for different groups of features: (**a**) error; (**b**) detection recall.

### 3.2. Maintenance Operations Detection Results

In this section, the classification results, using both median and mean filtered data, will be presented. Since this methodology will be used to validate the execution of maintenance operations in an FB, it must avoid the false positives. However, there are few true positive examples in the available data, and this is the reason why there is a thin balance between the recall and precision of each class. If the ANN is trained to detect all operations, it will decrease the precision of the classifier.

For the validation dataset, 10 ANN were generated with the previous specifications, to assess the sensitivity of the ANN classifier. The results are presented in Tables 6 and 7. The identification of maintenance operations is not possible in all cases. With the median filtered data, the detection results are worse in the validation dataset. Although, it is important to remember that in the validation dataset only six examples correspond to maintenance operations, so an error in just one example has a huge impact in these metrics. The precision of 57% for the detection of the operations is low, but is needed to guarantee that false positives rarely occur. Additionally, the recall value of 87% shows that most of the operations were detected.

**Table 6.** Classification results for the training dataset.

| | Median Filter Data | | Mean Filter Data | |
|---|---|---|---|---|
| **Detection** | **Yes** | **No** | **Yes** | **No** |
| Recall | 9% | 98% | 97% | 99% |
| Precision | 94% | 98% | 89% | 97% |
| F1-Score | 93% | 98% | 93% | 98% |
| Relative Error | 3.1% | | 3.3% | |

**Table 7.** Classification results for the validation dataset (average of all generated classifiers).

| | Median Filter Data | | Mean Filter Data | |
|---|---|---|---|---|
| **Detection** | **Yes** | **No** | **Yes** | **No** |
| Recall | 87% | 97% | 77% | 98% |
| Precision | 57% | 99% | 64% | 99% |
| F1-Score | 68% | 98% | 70% | 99% |
| Relative Error | 2.9% | | 2.5% | |

A more detailed analysis of the results for the validation dataset showed that the errors usually occur in continuous operations, just one of the months being identified. Evaluating the use of the mean filter, it can be verified that with the validation dataset, there is more precision in the detection (64%), i.e., there are less false detections of maintenance operations. The drawback is that the recall

decreases to 77%, being more difficult to detect the real operations. So, the median filter allows more operations detections, but due to its increased sensibility to transitions in the data, it produces more false detections than using the mean filter.

Both classifiers were applied to the real case scenario (Marisol) and correctly detected one maintenance operation in March 2018. Unfortunately, there were also two false detections in May and June of 2017. These can be explained with the fact that there was not any clear observation in May 2017 in Marisol, which led to more significant changes in the data in this period. The observations that confirm the results are in Figure 14.

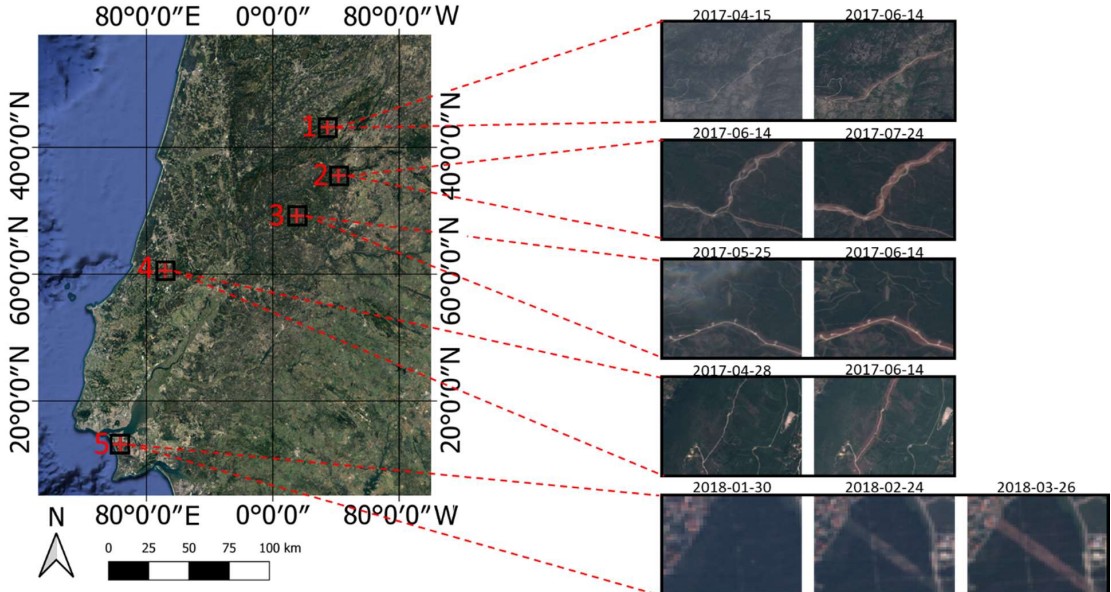

**Figure 14.** TCI observations before and after maintenance operations. 1-Seia; 2-Fundão; 3-Sertã; 4-Serra dos Candeeiros; 5-Marisol.

Finally, relative to the incomplete operations test dataset, the nine cases were divided in three groups of operation completeness. The results by group are presented in Table 8. (they were the same for both noise filters). Only one case was wrongly detected and it was in the 50%–75% group, where completeness of the operation on the FB was 73%. Additionally, in the tested FBs with a partial intervention of less than 50%, no operations were detected. This was a good indicator that incomplete operations will not be detected, which is important for the FBN monitoring.

**Table 8.** Results of the incomplete operations detection.

| Group | Number of Cases | Wrong Detections |
|---|---|---|
| 0%–25% | 3 | 0 |
| 25%–50% | 4 | 0 |
| 50%–75% | 2 | 1 |

## 4. Discussion

Monitoring of the forest is a fundamental task in fire prevention. Not only is it important to characterize the land cover, but also to understand what is changing and why it is happening, or to validate the execution of land management operations. As is known and verified during this study, the phenology is a major concern in change detection problems, since it represents transitions in the time-series without occurring any modification in the coverage. Additionally, when earth observation satellites such as Sentinel-2 are used, the variations in the luminosity add more background noise to the data. Another concern about using optical satellites is its sensitivity to adverse atmospheric

conditions, which leads to greater periods without available observations. Finally, a methodology to detect changes in any kind of forest needs to handle the different behaviour of vegetation types.

The results from the feature selection revealed that the better indices for the detection were based on the visible spectrum, namely ExR and ExR, which is understandable, since an operation is easily visible in TCI observations. Additionally, a further analysis of the time-series shows that these indices are more robust of the seasonal effects. This showed the importance of exploring other regions from the electromagnetic spectrum in remote sensing.

Compared with previous works, one of the achievements of this work was to achieve a monthly timestep for the change detection. This is important, since in [12,16,17], several observations were needed, and by using Landsat more than one month was needed to detect changes. In [13], annual composites were used, which only enabled the creation of annual change maps and in [15], the used observations were spaced by more than one month. The drawback was that the use of less data led to the occurrence of false positives (due to less confirmation steps) and the imposition of a month timestep obliged to estimate month values whenever there were no usable observations. In this work the error is approximately the same as in [12,13,15–17], but was tested in geographically separated areas, with different kinds of vegetation and during two full years, representing a more uncontrolled environment. In [10], the composites were generated in specific seasons of the year, avoiding the phenology effects. The proposed ANN was trained to detect only a specific kind of change, without discriminating changes as in [15–17].

The two steps for the noise reduction (object-based analysis and the application of noise filters) improved the quality of data and reduced false detections. Additionally, for filtering, two approaches were presented. The mean filter revealed more robustness to the false positives, although less ability to identify the desired events than the median filter. This is verified by the lower recall and higher precision of the mean approach. Although these methods helped to separate the phenological trends from the maintenance operations, misclassifications are still occurring. Since, in this case, it is very important to avoid the false positives, the mean filter was found to be a better solution.

The results in the test datasets were also promising, since in Marisol the expected operations were detected. Additionally, the results in the incomplete FBs test verified that this methodology could distinguish a complete operation from an incomplete. Although there is not a classification of the quality of the operation, if an operation is detected, it is known with good certainty that it was a correct operation.

## 5. Conclusions

The first and most important conclusion is the applicability of the presented methodology to the detection of maintenance operations in defined FBs. The results range between the 2.5% and 3.3% of error in the training and validation datasets. However, the F1-Scores in the validation dataset were in the order of 70%, which reveals that there still some false positives and false negatives. With the test dataset, the operations were only detected if more than 50% of the area was operated.

The wanted improvements, relative to other methods, were achieved; there is no need of confirmation if a change really happened; Sentinel-2 data, which give future works a better frequency of observations, were used; a specific kind of change is detected and usually is not confused with the background effects of phenology, and the less common vegetation indices had good performances in the detection.

As future work is expected to develop techniques for obtaining more observations, this compensates those affected by partial cloudy conditions. This implies the application of techniques to recover deteriorated images due to atmospheric conditions, and with the resource to radar data from Sentinel-1 (generating time-series of this data, as in [31]). The last may be used just for the image recovery, but also to get more information. Additionally, testing the ANN with the difference between the feature values in consecutive months instead of using the values of the two months is intended. This will lead to an ANN with less features; consequently, a simpler ANN. Finally, a different class for continuous

operations could be defined, that is characterized by smother changes, which can be misclassified as seasonal changes due to the phenology.

**Author Contributions:** Conceptualization, J.E.P.-P., V.A., A.M., J.M.N.S., J.M.F. and R.A.R.; methodology, J.E.P.-P. and A.M.; software, J.E.P.-P.; validation, J.E.P.-P., A.M., V.A. and J.M.N.S.; formal analysis, J.E.P.-P., V.A., A.M., J.M.N.S., J.M.F. and R.A.R.; investigation, J.E.P.-P., V.A., A.M., J.M.N.S., J.M.F. and R.A.R.; writing—original draft preparation, J.E.P.-P. and A.M.; writing—review and editing, J.E.P.-P., V.A., A.M., J.M.N.S., J.M.F. and R.A.R.; supervision, A.M. and J.M.N.S.; project administration, A.M., J.M.N.S., J.M.F. and R.A.R.; funding acquisition, A.M., J.M.N.S., J.M.F. and R.A.R. All authors have read and agreed to the published version of the manuscript.

**Funding:** This research was funded by Fundação de Ciências e Tecnologia (FCT) under the framework of projects FUELMON (PTDC/CCI-COM/30344/2017), foRESTER (PCIF/SSI/0102/2017) and PEST (UID/EEA/00066/2019). The Forest Research Centre is a research unit funded by FCT (UIDB/00239/2020).

**Acknowledgments:** The authors would like to acknowledge the Portuguese Institute of Nature and Forest Conservation (Instituto de Conservação da Natureza e das Florestas, ICNF) for presenting us this research topic and supplying data regarding the maintenance operations.

**Conflicts of Interest:** The authors declare no conflict of interest.

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
