# Peer review of "Semi-Automatic Methodology for Fire Break Maintenance Operations Detection with Sentinel-2 Imagery and Artificial Neural Network"

_remotesensing, doi:10.3390/rs12060909_

Round 1
Reviewer 1 Report
The paper presents a really interesting study. However, it is difficult to read. I suggest that authors consider restructuring the paper clearly separating methods and results. The discussion is weak and could be strengthened by having more discussion of your methods and results in relation to a lot more similar studies in the literature.
Figures need to be improved to become more legible.
Author Response
Dear Reviewer,
Thank you for your reading and interest in our manuscript and for the proposed modifications. They were considered and several changes applied leading to a clearer and more sustained document. A list of modifications is presented below.
Point 1: However, it is difficult to read. I suggest that authors consider restructuring the paper clearly separating methods and results. The discussion is weak and could be strengthened by having more discussion of your methods and results in relation to a lot more similar studies in the literature.
Response 1: As proposed the structure of the manuscript was modified to the following:
- Introduction;
- Materials and Methods;
- Results;
- Discussion;
- Conclusion.
Now, there is a clear definition between the methods and results. Also, a new section was written to discuss the results, compare with the state of art and highlighting the strengths of the proposed methodology. The state of art was also deepened, presenting more studies about change detection works. Relatively to the results, a new test dataset were added to a better evaluation of the methodology and its application for assessing the robustness to incomplete maintenance operations. Finally, corrections to the text were made to make the manuscript clearer and easier to read.
Point 2: Figures need to be improved to become more legible.
Response 2: Graphs were all changed to a format with better resolution and the figures improved.
Finally, all the detected grammar and spelling errors were corrected.
Reviewer 2 Report
The authors presented a methodology using Sentinel-2 images and ANN for fire break maintenance operation detection. The method presented has the advantages of automatic and low cost, and thus is important for forest fire protection. However, there are some issues that should be addressed before the manuscript can be accepted for publication.
Poor literature review. The manuscript is lack of literature support for relevant research.
Lack of detailed analysis and discussion on the research results. Dataset is crucial for training of neural network, the quantity of images used for training should be presented.
Line 83-86, means that each of the study areas were used in ANN classifier training.
But in line255-256, Marisol area as one of the study areas was tested and not used in any tanning stage?
Line 85, change “were” to “was”
Line 143-144 and Figure 5, where is “the dashed yellow ellipse” in figure 5b?
Line 191-193, the sentence “Since the time-series are noisy signals and a timestep was required for the change detection classifier, it was defined a monthly step which was considered enough for this application.” is relevant to the sentence “A monthly timestep was defined, because two months without observations is unusual in the chosen study areas. When there is not any clear observation for a month, the previous month values are used.” in Line 195-196. These two sentence should be integrated.
Line 195, the “high” in “a high median filter ” means what?
Line 222-223, change “output are” to “output is”
Line 395, change “tis” to “this”
Table 4, AGO do you mean AUG? SET is SEP?
Author Response
Dear Reviewer,
Thank you for your reading and interest in our manuscript and for the proposed modifications. They were considered and several changes applied leading to a clearer and more sustained document. A list of modifications is presented below.
Point 1: Poor literature review. The manuscript is lack of literature support for relevant research.
Response 1: The literature review was improved with several new articles presented in the Introduction, addressing their fundamental features. Also the novelty of our methodology was also better described.
Point 2: Lack of detailed analysis and discussion on the research results. Dataset is crucial for training of neural network, the quantity of images used for training should be presented.
Response 2: A dedicated section to the discussion was added to the manuscript. In this section a comparison with the state of art works was introduced. Also, a subsection relatively to Materials and Datasets was added giving a more detailed description of the dataset (the number of observations were presented, lines 194-196) and in the results section the number of samples (lines 353-354) using during the training stage was stated.
Point 3: Line 83-86, means that each of the study areas were used in ANN classifier training. But in line255-256, Marisol area as one of the study areas was tested and not used in any tanning stage?
Response 3: The data relatively to Marisol was not used for the ANN training, it was just a test area. This is now stated in lines 179-181.
Point 4: Line 191-193, the sentence “Since the time-series are noisy signals and a timestep was required for the change detection classifier, it was defined a monthly step which was considered enough for this application.” is relevant to the sentence “A monthly timestep was defined, because two months without observations is unusual in the chosen study areas. When there is not any clear observation for a month, the previous month values are used.” in Line 195-196. These two sentence should be integrated.
Response 4: The sentences in lines 191-193 and 195-196, were rewritten to the sentences 277-282.
Point 5: Line 195, the “high” in “a high median filter ” means what?
Response 5: The high median filter differs to the normal one when the number of elements of the set is even. In these cases, instead of calculating the mean of the two middle values, the higher value is used. This is now explained in lines 278-280.
Point 6: Table 4, AGO do you mean AUG? SET is SEP?
Response 6: You are correct. The table was changed.
Finally, all the detected grammar and spelling errors were corrected.
Reviewer 3 Report
The authors identify the support of maintenance of FireBreak (FB) network as a relevant task for sentinel 2 data. But the exact requirement of the task are unclear and not sufficiently detailed so that is difficult to understand if the example used in the training of ANN would produce a system able to produce relevant information for the stated goal.
Authors use as positive event to detect as a binary variable the maintenance operation over the background of the cyclical phenology of the vegetation and the trend of regrowth of vegetation were FB were performed. Given that variable is binary and example didn’t include bad operation the trained ANN do not guarantee to distinguish between a correctly maintained FB and incomplete operation so it cannot be used to evaluate operation quality. Trained ANN cannot be used either to identify the section of FB that need maintenance given that ANN was not trained with the image before maintenance as point to be detected. The only information that ANN can give is that maintenance was done without guarantee on the quality of operation. Is it sufficient to support operation?
Given that the scope of the training is unclear is difficult to evaluate the technical aspect of the article
Line 121 “despicable” is worst than bad while the authors seems to want to use “negligible” from the rest of the sentence.
line 248 present in [24], → described in [24],
line 262 “As can be seen in Error! Reference source not found.” → problem in reference please solve. other point in the text have lost references, watch out.
line 274 state in few word what approach was used before to use reference. In any case the standard deviation do not seems that is used as input to ANN, so maybe this information could be removed
Figure 9 and 10 are show to look at difference in phenology across station and between phenology and FB operation, respectively. Legend should report it
In any case these figures (9,10) are only exemplar given that real decision was not based on only two time series but on the overall dataset with the SelectKBest procedure. Maybe Figure 9 and 10 are redundant.
line 324-325 “Although they present similar results, groups 5 and6 are simpler because they use less features” In what sense they are simpler? Simpler to use? Less computation are needed? Less data training?
line 333 and 335 the author state that penalty is 8. It is not clear if it is a parameter that was changed in the cross-validation procedure or it was fixed. If it was fixed it would be best to state it in section 4.1 and not in 5.2. If penalty was changed authors should explicitly state it giving range and exploratory strategy.
line 395 spelling “tis” → “this”
line 396 “As future work is expected to develop techniques for obtaining more observations,” it is not clear, authors want to use other source of information than sentinel2 or use more feature extracted from sentinel2?
Author Response
Dear Reviewer,
Thank you for your reading and interest in our manuscript and for the proposed modifications. They were considered and several changes applied leading to a clearer and more sustained document, specially the one relatively to the test of incomplete operations. A list of modifications is presented below.
Point 1: But the exact requirement of the task are unclear and not sufficiently detailed so that is difficult to understand if the example used in the training of ANN would produce a system able to produce relevant information for the stated goal.
Response 1: The Introduction was changed to clearly present the goal of this work. We also added background to the state of art and described the improvements that this methodology adds relatively to the existing literature. Finally, the methodology requirements were presented.
Point 2: Authors use as positive event to detect as a binary variable the maintenance operation over the background of the cyclical phenology of the vegetation and the trend of regrowth of vegetation were FB were performed. Given that variable is binary and example didn’t include bad operation the trained ANN do not guarantee to distinguish between a correctly maintained FB and incomplete operation so it cannot be used to evaluate operation quality. Trained ANN cannot be used either to identify the section of FB that need maintenance given that ANN was not trained with the image before maintenance as point to be detected. The only information that ANN can give is that maintenance was done without guarantee on the quality of operation. Is it sufficient to support operation?
Response 2: The proposed methodology does not have the intention to classify the quality of the maintenance operations, just to detect if it is executed or not. The main purpose of the work is to help National Authorities to validate if a planned maintenance operation was effectively executed. Therefore, the methodology was designed to avoid the detection of incomplete maintenance operations.
To improve the Results section a new test dataset was added that only includes incomplete operations (with different execution levels). Since the information about incomplete operation did not exist, some observations were removed to simulate these circumstances. The method is explained in lines 182-188 and by Figure 7 and its results presented in lines 429-434 and Table 8.
Point 3: line 274 state in few word what approach was used before to use reference. In any case the standard deviation do not seems that is used as input to ANN, so maybe this information could be removed
Response 3: In line 274 is not the standard deviation but the relative standard deviation. This metric was used to validate if mean of the pixels was a good measure for the FB object. But there was an error in that sentence that may have generated this misunderstanding. It was stated this: “To validate if median value is a representative measure to describe a FB, the relative standard 273 deviation was calculated [25] and obtained 0.185.”, but it was not the median value, it was the mean value. The sentence was rewritten in lines 272-273.
Point 4: Figure 9 and 10 are show to look at difference in phenology across station and between phenology and FB operation, respectively. Legend should report it
In any case these figures (9,10) are only exemplar given that real decision was not based on only two time series but on the overall dataset with the SelectKBest procedure. Maybe Figure 9 and 10 are redundant.
Response 4: Indeed, the process of feature selection relied on the selectKbest method as you said. We agree that Figures 9 and 10 could be misplaced, being now presented in the Study Definition section. They are important to demonstrate how the phenology affects the methodology, and how it is important to distinguish operations from seasonal effects.
Point 5: line 324-325 “Although they present similar results, groups 5 and6 are simpler because they use less features” In what sense they are simpler? Simpler to use? Less computation are needed? Less data training?
Response 5: In lines 324-325 the meaning of simpler was usage of less feature that implied less computation and less neurons in the input layer of the ANN. This is now clearer in lines 381-384.
Point 6: line 333 and 335 the author state that penalty is 8. It is not clear if it is a parameter that was changed in the cross-validation procedure or it was fixed. If it was fixed it would be best to state it in section 4.1 and not in 5.2. If penalty was changed authors should explicitly state it giving range and exploratory strategy.
Response 6: The penalty definition procedure is now more clearly presented in lines 392-396.
Point 7: line 396 “As future work is expected to develop techniques for obtaining more observations,” it is not clear, authors want to use other source of information than sentinel2 or use more feature extracted from sentinel2?
Response 7: In line 396 we mean that we will study techniques that can fix partial cloudy images to generate more cloud-free images. One of the hypotheses is to combine Sentinel-2 with Radar data from Senitnel-1 (lines 486-490).
Finally, all the detected grammar and spelling errors were corrected.
Round 2
Reviewer 3 Report
Article in pdf is not a final version, correction are visible as well some comments, further some error of misspelling and strange omission ( see Major line 171). It is difficult to evaluate the paper, so the editor need to ensure that next version would be actually a clean version.
Major
The authors answered to requests in a satisfactory manner, but problem persist in the addition of a validation dataset of incomplete operation. It is not clear how the incomplete operation were divided in class of incompleteness and is peculiar that the last class has as boundary 100% completion and so is not distinguishable from complete dataset. In such a case is not clear why a detection would be considered a false positive given that completion of operation is indeed reached. Probably a graph with a continuous representation of the completion value (probably day of work over the total, authors should clarify) would help.
table 1 is not present in PDF, please take care to includes within it the formula for each index.
441 if the incompleteness data set is in the range of 75-100% of completion it means that it include also the complete. It would be relevant to know the actual higher value of completion of the incomplete dataset.
Minor
line 171 the number 2 is a table or a figure?
426 darwback -> drawback
Author Response
Dear Reviewer,
Thank you again for your comments, that leaded to a better document. The suggestions were taken into consideration and we hope that now the document is more understandable, clear and well sustained. We apologize for submitting the wrong pdf document into the MDPI platform.
Point 1: The authors answered to requests in a satisfactory manner, but problem persist in the addition of a validation dataset of incomplete operation. It is not clear how the incomplete operation were divided in class of incompleteness and is peculiar that the last class has as boundary 100% completion and so is not distinguishable from complete dataset. In such a case is not clear why a detection would be considered a false positive given that completion of operation is indeed reached. Probably a graph with a continuous representation of the completion value (probably day of work over the total, authors should clarify) would help.
Response 1: The completeness degree corresponds to the ratio between the area that have already been operated and the full area. Also, the samples in group 75%-100% were removed, when the coverage of the maintenance operation is greater than 75% it is considered a complete operation. In the previous version the operations were practically complete. It resulted in changes in lines 180-186 where it is explained. Also, Figure 7 was modified to clear this situation. In the Results section there were changes to, beginning with the number of samples that now are 9, divided into three completeness degree groups. Modifications were made in lines 424-427 and in Table 8.
Point 2: Table 1 is not present in PDF, please take care to includes within it the formula for each index.
Response 2: This was an exportation to PDF problem, that is sorted out. The Table contains all the indices.
Point 3: If the incompleteness data set is in the range of 75-100% of completion it means that it includes also the complete. It would be relevant to know the actual higher value of completion of the incomplete dataset.
Response 3: Indeed, that is right, it was a mistake in the dataset. Those samples were removed, and Table 8 and lines 424-427 were updated.
Point 4: line 171 the number 2 is a table or a figure? 426 darwback -> drawback
Response 4: Correct, it corresponds to a Table; it is now revised in line 156.
We hope to have achieved a better manuscript, to have clearly explained part of the document and to have corrected everything as you expected.